# Serratiopeptidase Affects the Physiology of *Pseudomonas aeruginosa* Isolates from Cystic Fibrosis Patients

**DOI:** 10.3390/ijms232012645

**Published:** 2022-10-20

**Authors:** Marco Artini, Gianluca Vrenna, Marika Trecca, Vanessa Tuccio Guarna Assanti, Ersilia Vita Fiscarelli, Rosanna Papa, Laura Selan

**Affiliations:** 1Department of Public Health and Infectious Diseases, Sapienza University, p.le Aldo Moro 5, 00185 Rome, Italy; 2Research Unit of Diagnostical and Management Innovations, Children’s Hospital and Institute Research Bambino Gesù, 00165 Rome, Italy

**Keywords:** virulence, *Pseudomonas aeruginosa*, biofilm, cystic fibrosis, invasion, serratiopeptidase, antibiotic resistance, pyocyanin

## Abstract

*Pseudomonas aeruginosa* is frequently involved in cystic fibrosis (CF) airway infections. Biofilm, motility, production of toxins and the invasion of host cells are different factors that increase *P. aeruginosa’s* virulence. The sessile phenotype offers protection to bacterial cells and resistance to antimicrobials and host immune attacks. Motility also contributes to bacterial colonization of surfaces and, consequently, to biofilm formation. Furthermore, the ability to adhere is the prelude for the internalization into lung cells, a common immune evasion mechanism used by most intracellular bacteria, such as *P. aeruginosa*. In previous studies we evaluated the activity of metalloprotease serratiopeptidase (SPEP) in impairing virulence-related properties in Gram-positive bacteria. This work aimed to investigate SPEP’s effects on different physiological aspects related to the virulence of *P. aeruginosa* isolated from CF patients, such as biofilm production, pyoverdine and pyocyanin production and invasion in alveolar epithelial cells. Obtained results showed that SPEP was able to impair the attachment to inert surfaces as well as adhesion/invasion of eukaryotic cells. Conversely, SPEP’s effect on pyocyanin and pyoverdine production was strongly strain-dependent, with an increase and/or a decrease of their production. Moreover, SPEP seemed to increase swarming motility and staphylolytic protease production. Our results suggest that a large number of clinical strains should be studied in-depth before drawing definitive conclusions. Why different strains sometimes react in opposing ways to a specific treatment is of great interest and will be the object of future studies. Therefore, SPEP affects *P. aeruginosa’s* physiology by differently acting on several bacterial factors related to its virulence.

## 1. Introduction

Cystic fibrosis (CF) is a genetic disease, where a mutation of a protein called cystic fibrosis transmembrane conductance regulator (CFTR) leads to an alteration of ion transport across the cellular membrane and the to the production of thick, sticky mucus that clogs the airways and traps microorganisms [1]. This condition leads to inflammation and persistent bacterial infections, which cause principally respiratory malfunction [2]. The lung of young CF patients is primary colonized by *Haemophilus influenzae* and *Staphylococcus aureus* and then by the opportunistic pathogen *Pseudomonas aeruginosa* [3,4].

Recently, the Infectious Diseases Society of America focused the attention on a fraction of antibiotic-resistant bacteria, acronymically named ‘the ESKAPE pathogens’, capable of ‘escaping’ the biocidal action of antibiotics and mutually representing new paradigms in pathogenesis, transmission and resistance [5]. In this category is included the versatile opportunistic pathogen *P. aeruginosa,* responsible for both acute and chronic infections. It possesses a wide and variable collection of virulence factors and antibiotic resistance determinants, that enable it to adapt to multiple conditions and survive in hostile environments, such as those that occur in the pulmonary tract of CF patients. The lung environment is extremely aggressive for the presence of elevated osmotic stress, high concentrations of antimicrobial drugs, reduced nutrient availability and interspecies competition. This latter forces *P. aeruginosa* to adapt for its survival and to adopt peculiar strategies to escape the host immune response [6]. The physiological adaptability and flexibility of this pathogen enable it to promote the development of highly complex chronic infections.

Actually, inhibition of virulence rather than bacterial viability is considered a novel promising strategy for the development of antipathogenic agents [7,8,9,10,11,12], since molecules acting on specific bacterial virulence programs should not induce the selection of resistant strains observed after antibiotic treatment [13]. Anti-virulence strategies undoubtedly exert a lower selective pressure as compared to antimicrobial drugs. However, in order to target specific virulence factors, a deep knowledge of the physiology of the pathogen is necessary, including the consequences of the relationship with the host. In fact, pathogens adapt specific virulence programs to the environmental conditions of the host colonized districts at a given moment. Therefore, the identification of a chemical entity promoting physiological changes of a pathogen related to specific virulence factors is certainly complicated.

The anti-virulence strategy is particularly suitable for chronic infections related to multidrug-resistant bacteria, such as those responsible for lung recurrent/chronic infections associated with cystic fibrosis (CF) disease.

Enzymes directly or indirectly control many physiological functions in the human body. Actually, enzymes derived from different origins such as plants, animals, and microbial sources are used in clinical practice for the treatment and management of various diseases [14]. Among these are trypsin, chymotrypsin, papain and bromelain, whose therapeutic efficacy is widely demonstrated [15].

In particular, in the field of infectious diseases, due to the appearance of multidrug-resistant strains and to the inexorable failure of conventional treatments, enzyme-based therapeutics play an important role for their high specificity of action. The therapeutic efficacy of various enzymes including hydrolases, and more specifically proteases, has been recognized [14]. Indeed, proteases, in particular metalloproteases, were hypothesized to represent a potential treatment for infections sustained by bacterial biofilms [16]. Moreover, several commercial proteases have shown to be effective in the eradication of biofilms [17,18].

Among proteases, serratiopeptidase, commonly defined SPEP, is a proteolytic enzyme widely used in therapeutic applications. SPEP is a metalloprotease of bacterial origin with well-known anti-inflammatory, anti-edemic and analgesic effects, largely used for its proteolytic and fibrinolytic properties in various areas including surgery, gynaecology, otorhinolaryngology, orthopaedics, and dentistry [15,19]. Other applications include sinusitis, breast disease and lung disorders [20]. Recently, the use of SPEP in therapy, alone or in combination with other drugs, has globally increased.

In the field of infectious diseases, SPEP is recognized to be effective against a variety of biofilm-associated chronic infections and against adhesion and invasion of bacteria on eukaryotic cells [21,22,23,24].

It is worthy to note that biofilm phenotype is usually associated with recurrent and chronic infections that are particularly difficult to eradicate. SPEP was shown to enhance ofloxacin activity on sessile cultures of *P. aeruginosa* and *S. epidermidis* [25]. On *S. aureus*, we also evaluated the effect of SPEP on attachment to inert surfaces and adhesion/invasion on eukaryotic cells. Results obtained showed that although SPEP did not affect bacterial vitality, it impaired the adhesion to abiotic surfaces, affecting biofilm formation and invasiveness by *S. aureus* to human cells [23].

An important virulence factor expressed by *P. aeruginosa* resides in its ability to form biofilm that facilitates the attachment to biotic or abiotic surfaces and confers a high resistance to host immune response and a high tolerance to elevated concentrations of antimicrobial agents. Biofilm colonization of a biotic surface such as an epithelium, can represent the prelude to the invasion of the host eukaryotic cells. Invasiveness too represents a form of phenotypic resistance to antibiotics.

Further virulence factors of *P. aeruginosa* are represented by exoproducts; pyoverdine, the main siderophore of *P. aeruginosa*, is involved in the regulation of multiple bacterial virulence factors, such as exotoxin A and endoprotease, and is crucial for the uptake of extracellular iron due to its high affinity for Fe^3+^, which is an essential nutrient for biofilm formation. Pyocyanin is a zwitter-ion that can easily penetrate biological membranes. Finally, pyocyanin production induces oxidative stress in host tissues, disrupting host catalase, and mitochondrial electron transport [26,27,28,29]. For these reasons it is responsible for the severity of the disease.

Furthermore, the zinc-dependent metalloprotease LasA is required for maximal expression of the elastinolytic phenotype of *P. aeruginosa*. By cleaving peptide bonds within the pentaglycine bridges that stabilize the cell wall peptidoglycan of *Staphylococci*, staphylolysin can lyse and kill *S. aureus* cells, Since *P. aeruginosa* infections are often secondary to infections with *S. aureus*, staphylolysin may also play an important role at the colonization stage of the infection, when the elimination of competing organisms such as *S. aureus* would be an advantage [30].

This work aimed at a deep investigation of SPEP’s capability to interfere with physiological aspects of *P. aeruginosa* clinical isolates obtained from CF patients connected with its virulence, such as biofilm formation and accumulation, pyocyanin and pyoverdine production, swarming and swimming motility, staphylolytic activity and adhesion/invasion on eukaryotic cells.

## 2. Results

### 2.1. Effect of SPEP on Biofilm Formation

First, antimicrobial activity of SPEP was assessed on clinical and reference *P. aeruginosa* strains. Results obtained showed that SPEP did not affect bacterial vitality up to a concentration of 2000 U/mL (Appendix A). Preliminary experiments were performed to define the optimal SPEP concentration for the assessment of the anti-biofilm response. For these experiments, the reference strain *P. aeruginosa* PA14 was used. This strain is a highly virulent and biofilm hyperproducer characterized by a specific mutation in ladS gene [31]. Figure 1 reports the SPEP effect on PA14 biofilm formation, starting from a concentration of 2000 U/mL (row data are presented in Appendix A). SPEP was active in reducing biofilm formation even at relatively low concentrations, however its activity did not show a dose-dependent trend.

At lower concentrations, SPEP retains a partial antibiofilm activity. Based on these results, the adopted SPEP concentration for further experiments was 200 U/mL, because it assures about 50% biofilm disaggregation. Afterwards, SPEP activity was evaluated on clinical strains of *P. aeruginosa*. SPEP was added to the medium at the beginning of the cultivation (0 h, biofilm formation), or after 24 h of bacterial culture (24 h, mature biofilm). As a control, bacteria were cultured in BHI medium without SPEP. The effects of SPEP on biofilm formation are presented in Figure 2, Panel A. Results are expressed as a ratio between biofilm formed in presence of SPEP and biofilm produced by bacteria grown only in BHI medium.

SPEP showed an anti-biofilm activity on three out of four tested clinical strains, with a percentage of inhibition ranging between 70% and 40%. The strongest inhibition (70%) was achieved on the isolate 28P. Raw data are presented in Appendix A, panel A.

SPEP effect was also assessed on a 24 h mature biofilm. Bacterial strains were grown in microtiter plates at 37 °C in BHI. After 24 h, the growth medium was gently removed and replaced with BHI containing or not containing 200 U/mL SPEP. The incubation was performed for a further 24 h at 37 °C. After the addition of SPEP, in three out of four clinical strains biofilm reduction was observed, ranging from 20 to 40% (Figure 2, panel B). As reported in Table 1, the biofilm after 48 h was much more abundant and probably more structured and difficult to eradicate, except for the 23P strain, where a lower biofilm biomass was observed at 48 h. These data are not surprising, being that biofilm kinetics are normally characterized by alternating phases of biofilm development and cell detachment [32]. It is noteworthy to observe the effect of SPEP on the biofilm hyperproducer PA14, where the molecule was able to disaggregate about 60% of biofilm. Raw data are presented in Appendix A, panel B.

This result suggests that SPEP action is not limited to initial bacterial adhesion to the substrate but is also effective on preformed and highly structured biofilms.

Furthermore, we evaluated the effects of SPEP on the growth curve of *P. aeruginosa* strains that were more influenced by its action. SPEP was added to the medium at the beginning of the growth and bacterial curves were monitored for over 30 h. As a control, bacteria were grown in the absence of SPEP. SPEP did not impair the *P. aeruginosa* duplication rate of tested strains as reported in Figure 3. Indeed, bacterial growth curves were approximately superimposable both in the presence or absence of SPEP.

### 2.2. Effect of SPEP on Pyocyanin and Pyoverdine Production

SPEP’s influence on pyocyanin and pyoverdine production was assessed on clinical strains and the reference strain *P. aeruginosa* PA14.

First, we investigated the effects of SPEP on the expression and production of the virulence-associated pigment pyocyanin. SPEP treatment was performed on bacteria cultured for 48 h and pyocyanin production was evaluated at 24 and 48 h. As a control, bacteria were cultured in BHI medium in the absence of SPEP. For each strain, data are reported as a percentage of pyocyanin production after 24 and 48 h of treatment, in comparison with that of the untreated sample (Figure 4). Pyocyanin production was normalized to the optical density reached by the bacterial culture at each time point.

SPEP seemed to slightly reduce pyocyanin production in three out of five tested strains after 24 h of treatment. The reduction observed was statistically significant only for the strain 27P (reduction of 35.5% compared to the control). Instead, after 48 h a slight significative increase was observed for strains 23P, 27P and 28P. In particular, for strain 28P a significantly higher production of pigment was observed following SPEP treatment. Raw data are presented in Appendix A.

Lastly the effect of SPEP on the main siderophore, pyoverdine, of *P. aeruginosa* was evaluated. Pyoverdine production was spectrophotometrically evaluated at two different time points during the growth curve, at 24 and 48 h. Results obtained showed that it accumulated in the growth medium in four out of five strains only in the late stationary phase (48 h). Indeed, as reported in Table 2 (Section 4), pyoverdine was not produced in our experimental condition by strain 23P. Consequently, the effect of SPEP on pyoverdine production was only evaluated on strains PA14, 27P, 28P and 30P at 48 h of growth. Results are reported in Figure 5 (raw data are represented in Appendix A).

Particularly interesting are the results obtained for the ATCC PA14 and 27P clinical strains: a complete inhibition was observed for the reference strain PA14, while for 27P strain a dramatic reduction in pyoverdine production after SPEP addition was evidenced, approximately corresponding to 95.7% (4.3% of residual pyoverdine).

### 2.3. Effect of SPEP on Motility of P. aeruginosa

The ability of SPEP to interfere with swarming and swimming motility patterns in clinical *P. aeruginosa* isolates was also assayed. Swarming is defined as a fast, coordinated movement of bacteria on a semi-solid surface. SPEP effect was tested at 200 U/mL, as reported in Figure 6, left panel. SPEP was able to increase swarming motility in all tested strains (Figure 6). This effect was less pronounced for reference strain PA14. The swimming of bacteria in liquid environments is enabled by polar flagella movements. The presence of the SPEP did not modify swimming motility in all the tested strains.

### 2.4. Effect of SPEP on Staphylolytic Activity of P. aeruginosa

The ability of SPEP to interfere with LasA production was performed using the disk diffusion susceptibility assay of *S. aureus* growth inhibition by supernatant of *P. aeruginosa* treated and untreated with SPEP (Figure 7). This assay is used to assess the susceptibility of *S. aureus* strains to staphylolysin. *S. aureus* cells are spread evenly on a Mueller-Hinton agar plate. Sterile filter paper disks impregnated with free culture supernatants of *P. aeruginosa* strains grown in presence or in absence of 200 U/mL of SPEP are placed on the agar surface; the plates are incubated overnight at 37 °C to permit bacterial growth. As shown in Figure 7, *P. aeruginosa* supernatants did not inhibit staphylococcal growth. An inhibition of growth was observed only in the presence of the supernatant derived from the 23P strain grown in the presence of SPEP.

### 2.5. Effect of SPEP on Adhesion and Invasion of P. aeruginosa to Eukaryotic Cells

SPEP action was also tested on *P. aeruginosa’s* ability to adhere and invade human cells. Since the bacterial strains studied in this paper have been isolated from CF patients with chronic airway infections, adenocarcinomic human alveolar basal epithelial cells A549 were used for this experiment.

First, bacterial strains with resistance to gentamicin were evaluated. Data obtained showed that strains 23P, 28P and 30P were resistant to this antibiotic up to a concentration of 300 µg/mL. For this reason, the effect of SPEP on adhesion and invasion of eukaryotic cells was evaluated only on the reference strain *P. aeruginosa* PA14 and on the clinical strain 27P. Absence of gentamicin toxicity toward A549 cells was assessed too (Appendix A). Taking into account the high adhesion ability of *P. aeruginosa* to eukaryotic cells, a multiplicity of infection (MOI) of 1:100 was used. Adhesion and invasion of SPEP-treated and untreated bacteria are reported in Table 1. Adhesion was defined by the number of bacteria adherent on A549 cells after 1 h of incubation. Invasion represents the number of internalized bacteria that survived after the lysis of A549 cells by gentamicin (1 h incubation on cells plus an additional hour for gentamicin treatment). The adhesion efficiency of *P. aeruginosa* clinical and reference strains is in the same order of magnitude, corresponding to about 10% of total CFU used (about 105 bacteria). Furthermore, our results showed that the adhesion efficiency of both *P. aeruginosa* strains was unaffected by SPEP incubation. These data are statistically significant (*p* value < 0.05). As regards the invasion, our data showed that about 1% of total bacteria adhering to A549 cells are able to invade them (about 103 bacteria, *p* value < 0.05). Interestingly the invasion efficiency was significatively reduced following SPEP treatment in both analyzed strains (more than 100-fold, *p* value < 0.05).

## 3. Discussion

This study focused on the modulation applied by SPEP on the physiology of *P. aeruginosa* strains isolated from CF patients, with particular attention paid to the modulation of virulence factors.

It is important to consider that bacterial strains isolated from different CF patients behave differently from each other, because the prolonged conflict with other commensals (bacteria and mycetes) and with the unique profile of the immune response of the single patient compels bacteria to adapt and change survival strategies within the host, molding their physiology according to the environmental conditions.

In addition, the expression of phenotypic profiles associated with the establishment of chronic infections converts this opportunistic pathogen into an extraordinary competitor for the lung environment. *P. aeruginosa* is the dominant pathogen that persists in individuals with CF up to the end of their life, since it adapts its physiological profile by decreasing the expression of virulence factors that usually are detected by the immune system of the host [2]. Thanks to a wide array of virulence factors, including determinants for antibiotic resistance, *P. aeruginosa* can survive in hostile environments, developing highly complex chronic infections.

*P. aeruginosa* strains are responsible for chronic infections, especially if associated with CF, lose the ability to move, thus escaping the host immune response. Once bacteria colonize the host airways, they develop chronic infections, mainly producing biofilm [2]. The ability to form biofilm, a community of microorganisms embedded within a matrix of self-produced polymeric material, allows bacterial cells to elude the immune response and obtain protection from antimicrobials. Biofilm formation, invasion of the host cells and production of exotoxins represent important weapons that *P. aeruginosa* can variably and alternatively express during its residency in the lungs of CF patients. In chronic infection, *P. aeruginosa* release cyanine and rhamnolipids with a gradual decrease in their motility [33]. 

Taking into account what is aforementioned, novel strategies to defeat *P. aeruginosa* infection, especially during its persistence in the CF airway, are necessary.

Under this point of view, it’s interesting to understand the effect of SPEP on some physiological aspects of clinical *P. aeruginosa* strains that are so relevant for the development of chronic infection. Furthermore, the demonstrated ability of SPEP to interfere with the adhesive properties of bacterial pathogens [21,22,23,24] drives researchers to further investigate its action on other virulence factors.

In our experiments, SPEP appears to increase the swarming motility of *P. aeruginosa* clinical strains, leaving an unaltered swimming-type motility. The effect of SPEP, therefore, on one hand facilitates biofilm breakdown and on the other hand restores bacterial motility.

It is very difficult, probably impossible, to find a single compound able to completely knock down bacterial virulence in all tested clinical strains [34,35], probably due to phenotypic diversification induced by adaptation to the environment of different patients. This limit in anti-virulence activity exerted only on some tested strains also affects this study. On the other hand, we think that it is worthy to keep on in the search of new compounds that modulate bacterial physiology so as to reduce virulence, even if they can result in being inactive on some strains. In fact, only when a wide array of partially active compounds are collected, will be possible to design blends of anti-virulence compounds acting in synergy in nearly all *P. aeruginosa* strains.

On the other hand, it is worthy to remind ourselves that the weakening of bacterial virulence, albeit diversified from strain to strain (meaning also from patient to patient) can allow the reinforcement of the immune defense and restore its efficacy, obtaining the overturning of forces in favour of the host. This aim, in parallel with the avoidance of further selection of antibiotic resistances, can lead to a better control of the chronic infections that pave the path to the fatal progression of CF patients to the end.

In general terms, as a starting point, the administration of SPEP (and of any other innovative drug targeting virulence features rather than the vitality of bacteria) could be first evaluated as an ad hoc therapy in patients where the multidrug resistance of bacteria determines the failure of medical therapies.

Our results, sometime divergent depending on the investigated clinical isolate, suggest that a large number of clinical strains should be in-depth studied before drawing definitive conclusions. Why different strains sometimes react in opposing ways to a specific treatment is of great interest and will be the object of future studies.

In conclusion, our results showed that physiological changes induced by SPEP in *P. aeruginosa* could impact some aspects of the pathogenesis during CF infections.

## 4. Materials and Methods

### 4.1. Ethics Approval and Informed Consent

The ethics committee of the Pediatric Hospital and Institute of Research Bambino Gesù (OPBG) in Rome, Italy approved the following research (No. 1437_OPBG_2017 of July 2017). This study was conducted in respect of the Declaration of Helsinki as a statement of ethical principles for medical research involving human subjects. All participants and legal guardians signed an informed consent form.

### 4.2. Chemicals

Serratiopeptidase enzyme (2540 U/mg) was kindly supplied by Takeda Italia Farmaceutici (Rome, Italy). The stock solution was obtained by dissolving the powder in PBS at a concentration of 20,000 U/mL and stored at −20 °C until use.

### 4.3. Bacterial Strains and Growth Conditions

Clinical strains of *P. aeruginosa* were isolated from respiratory specimens from CF patients in follow-up to OPBG. Phenotypic characteristics of the bacterial strains are summarized in Table 2. Antimicrobial profiles of *P. aeruginosa* strains are summarized in Appendix A [36]. The reference strain used was *P. aeruginosa* PA14 [31]. Bacteria were grown in Brain Heart Infusion broth (BHI, Oxoid, Basingstoke, UK). For pyoverdine quantification, bacteria were grown in King’s B Medium supplemented with 0.5% (*w/v*) of Casamino Acid (CAA) [37]. Bacterial cells were grown in planktonic conditions at 37 °C under orbital shaking (180 rpm), while the biofilm formation was performed under static conditions.

Growth curves were performed at 37 °C under orbital shaking (180 rpm) in BHI medium in the presence and absence of 200 U/mL of SPEP. Overnight bacterial cultures were inoculated in fresh medium starting from OD 600 nm about 0.06. The cultures were spectrophotometrically monitored at 600 OD each hour for about 24 h.

### 4.4. Biofilm Formation

The biofilm content was quantified by a microtiter plate (MTP) biofilm assay [38]. A total of 0.5 OD 600 nm of bacterial suspension in the exponential growth phase was 1:100 diluted into the wells of a sterile 96-well polystyrene flat base plate prefilled with BHI medium in the presence and in the absence of SPEP, as previously reported [23]. The plates were incubated at 37 °C under static conditions overnight. After incubation, the supernatant containing planktonic cells was gently removed by inverting the plate and washing the microtiter plate three times with double-distilled water. Then, the microtiter plate was patted dry in an inverted position. The staining was performed with 0.1% crystal violet for 15 min at room temperature. The excess of crystal violet was removed rinsing by twice with double-distilled water, and thoroughly dried to quantify the biofilm formation. The adherent biofilm was solubilized with 20% (*v/v*) glacial acetic acid and 80% (*v/v*) ethanol, and spectrophotometrically quantified at 590 nm. Each data point is composed of four independent experiments, each performed in at least three replicates.

### 4.5. Mature Biofilm

Assays were also performed on preformed biofilms [39]. A total of 100 µL of BHI medium containing 1:100 dilution of overnight bacterial culture was used to fill the wells of a sterile 96-well flat-bottomed polystyrene plate and incubated for 24 h at 37 °C under static conditions. Then, the content of the plates was poured off and the wells were washed to remove the unattached bacteria; 100 μL of fresh BHI containing SPEP was added into each well. As control, 100 μL of fresh medium was added to the first row of the microtiter plate. The plates were incubated for additional 24 h (48 h total) at 37 °C. After 24 h the plates were analyzed as previously described.

### 4.6. Pyocyanin Assay

The determination of pyocyanin production was performed as previously described [39]. Briefly, single bacterial colonies were inoculated separately in 10 mL of BHI broth with or without 200 U/mL of SPEP and incubated for different times at 37 °C under 180 rpm orbital shaking. The saturated suspension was centrifuged at 10,000 rpm for 15 min, the cell-free supernatant was used for the extraction of the pyocyanin. Chloroform was added to the supernatant in a ratio of 1:1. The solution was mixed by inversion and then decanted for 15 min to allow the separation between the two phases. The lower layer containing pyocyanin was transferred into a tube containing 0.2 M HCl. It was mixed and decanted to let the separation of the two phases. The pink-colored upper layer containing pyocyanin was recovered and spectrophotometrically quantified at 520 nm. Pyocyanin was normalized for the optical density detected in each bacterial culture.

### 4.7. Pyoverdine Assay

For pyoverdine quantification, a total of 100 μL of each *P. aeruginosa* cell-free supernatant was added to a black 96-well plate (Greiner, Stonehouse, UK) and read at excitation and emission wavelengths of 400/460 nm as previously reported [40] on an Infinite 200 PRO (Molecular Devices, San Jose, CA, USA) fluorescence microplate reader. The background level of fluorescence was measured using 100 μL King’s B medium supplemented with 0.5% (*w/v*) CAA broth and subtracted from the sample measures. Each data point consists of three independent experiments, each performed in at least four replicates.

### 4.8. Motility Assays

#### 4.8.1. Swarming Assay

The swarming assay was performed as previously published by Vrenna and coworkers [39], with some modifications. Briefly, 200 U/mL SPEP was added to molten swarming agar at a concentration of 200 U/mL. Swarming agar was prepared as follows: 0.8% Nutrient Broth (Oxoid, Basingstoke, UK), 0.5% D-(+)-glucose (Sigma, Steinheim, Germany) and 0.5% agarose (Invitrogen, Paisley, UK). The culture was then dispensed onto Petri dishes after gentle mixing. Once the culture was solidified, 2 μL of each overnight *P. aeruginosa* culture, grown respectively in the presence and absence of SPEP, was inoculated in the center of the agar and then incubated at 37 °C for 24 h. As control, we used molten swarming agar without SPEP. After the incubation period, diameters of the growth zones were measured. 

#### 4.8.2. Swimming Assay

The swimming assay was conducted according to previous research [41], with some modifications. The procedures were the same as those of the swarming assay, except for the swimming agar composition, which consisted of 1.0% peptone (Oxoid, Basingstoke, UK), 0.5% sodium chloride (Sigma, Steinheim, Germany), and 0.3% Bacto-Agar (BD, Le Pont de Claix, France). After the incubation period, diameters of the growth zones were measured.

### 4.9. Disk Diffusion Staphylolysin Susceptibility Assay

For this experiment, two reference strains of *S. aureus* were used: *S. aureus* ATCC 25,923 and *S. aureus* ATCC 6538P.

Several colonies of *S. aureus* were collected from a BHI fresh plate and transferred to a sterile test tube containing 3 mL of sterile PBS using a sterile swab. The cell suspension was gently mixed until a homogenous solution was formed. The absorbance of the suspension at 600 nm was adjusted to ~0.6 OD units. Bacterial cell suspension was homogeneously spread onto a Mueller-Hinton agar plate. Cells were adsorbed to the surface of the agar for 30 min at room temperature. Sterile filter paper disks were carefully applied to the agar’s surface and impregnated with 20 μL of free culture supernatants of *P. aeruginosa* strains grown in the presence and absence of 200 U/mL SPEP, respectively. The plates were incubated overnight at 37 °C. As a control, a disk impregnated only with 200 U/mL SPEP was used. The width of the clear zone provides a measure of the relative staphylolytic activity.

### 4.10. Eukaryotic Cells

The adenocarcinomic human alveolar basal epithelial cells A549 were cultured in minimal essential medium with Earle’s balanced salt solution and high glucose at 4.5 g/L (MEM/EBSS), supplemented with 10% fetal calf serum (FCS), 1% glutamine and 1% penicillin–streptomycin in an atmosphere of 5% CO_2_ at 37 °C. All media were from Euroclone (Milan, Italy). Confluent monolayers were used 24 h after seeding.

### 4.11. Gentamicin Toxicity Assay on A549 Cells 

The cytotoxicity of gentamacin to adenocarcinomic human alveolar basal epithelial cells (A549) was assessed using an MTT cell proliferation kit (Roche Applied Science, Penzberg, Germany). A549 cells were incubated with 200 and 300 µg/mL of gentamicin at 37 °C under 5% CO_2_ for 1 and 2 h. A 50 μL volume of MTT working solution was added to each well, and the mixture was incubated for other 4 h. The purple crystal formazan was observed around cells at ×40 magnification under a microscope. The cell medium was carefully removed, and then 100 μL of dimethyl sulfoxide (DMSO) was added to each well to dissolve formazan. After 15 min of incubation at 37 °C to completely dissolve formazan, the absorbance at 490 nm was measured on Tecan Infinite 200pro microplate (Tecan Group Ltd., Männedorf, Switzerland). Cell survival was expressed as a percentage of viable cells in the presence of gentamicin, with respect to control cells. Control cells are represented by cells grown in the absence of the molecule supplemented with identical volumes of DMSO. The Student’s *t*-test was performed for statistical analysis.

### 4.12. Antibiotic Protection Assay

Strains of *P. aeruginosa* were cultured overnight in BHI broth at 37 °C at 180 rpm. Then, bacteria were diluted to 1:100 in prewarmed BHI and sub-cultured up to OD 600 = 0.5 at 37 °C with or without 200 U/mL of SPEP (SPEP-treated and non-SPEP-treated bacteria, respectively). Before infection, semi-confluent monolayers (1.25 × 105 cells/well) of human epithelial cells (A549) were cultured in 24-well plates (BD Falcon, USA) in basal medium containing 10% FCS with 1% antibiotic (penicillin–streptomycin) at 37 °C and 5% CO_2_ to obtain a confluent monolayer after 24 h. At least one hour prior to infection of the human cells, the culture medium was replaced with DMEM medium plus 1% glutamine added with 2% CSF without antibiotics. Then, human cells were separately infected with SPEP-treated and SPEP-untreated bacterial suspension at a multiplicity of infection (MOI) of about 10 bacteria per cell (MOI 10:1) for 1 h at 37 °C and 5% CO_2_.

The adhesion test was performed by keeping cells and bacteria in contact for 1 h at 37 °C. Loosely bound bacteria were removed from cell monolayers by two washes with PBS. The cells were then lysed with 0.025% Triton X100, serially diluted and plated on MacConkey agar (Oxoid, Basingstoke, UK) to determine viable adherent and internalized bacteria.

To count the internalized bacteria, epithelial cell monolayers were washed with PBS after incubation, and 0.5 mL of fresh medium containing 300 µg/mL of gentamicin was added to each well and incubated again for 1 h at 37 °C and 5% CO_2_ to kill extracellular bacteria. After this additional hour, cells were lysed with 0.025% Triton X-100 and the collected supernatants were plated on MacConkey agar (Oxoid, Basingstoke, UK) followed by an overnight incubation at 37 °C to count viable intracellular bacteria. The internalization efficiency is expressed as a percentage of the inoculated bacteria. Data represent the mean of three independent experiments. Adhesion efficiency was expressed as the percentage of the inoculated bacteria that adhered to A549 cells.

## Figures and Tables

**Figure 1 ijms-23-12645-f001:**
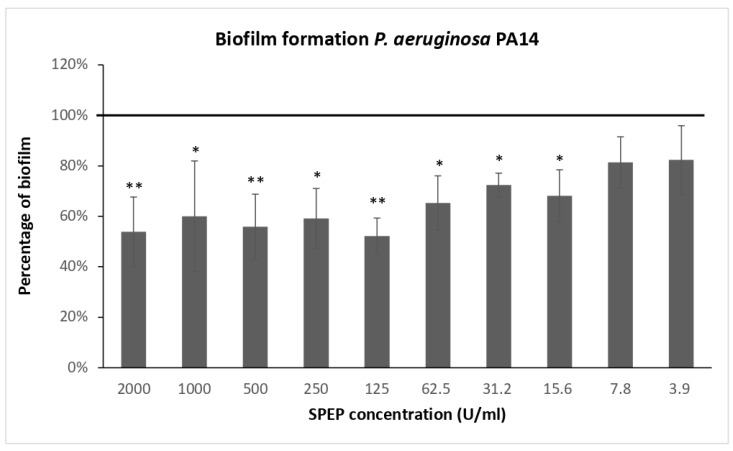
SPEP effect on *P. aeruginosa* PA14 biofilm formation starting from a concentration of 2000 U/mL. Results are expressed as the percentage of residual biofilm formation compared with an untreated sample. Data reported are representative of three independent experiments. Error bars indicate the standard deviations of three measurements. Statistical differences were determined by the Student’s *t*-test: * *p* < 0.05; ** *p* < 0.01, compared with the untreated samples.

**Figure 2 ijms-23-12645-f002:**
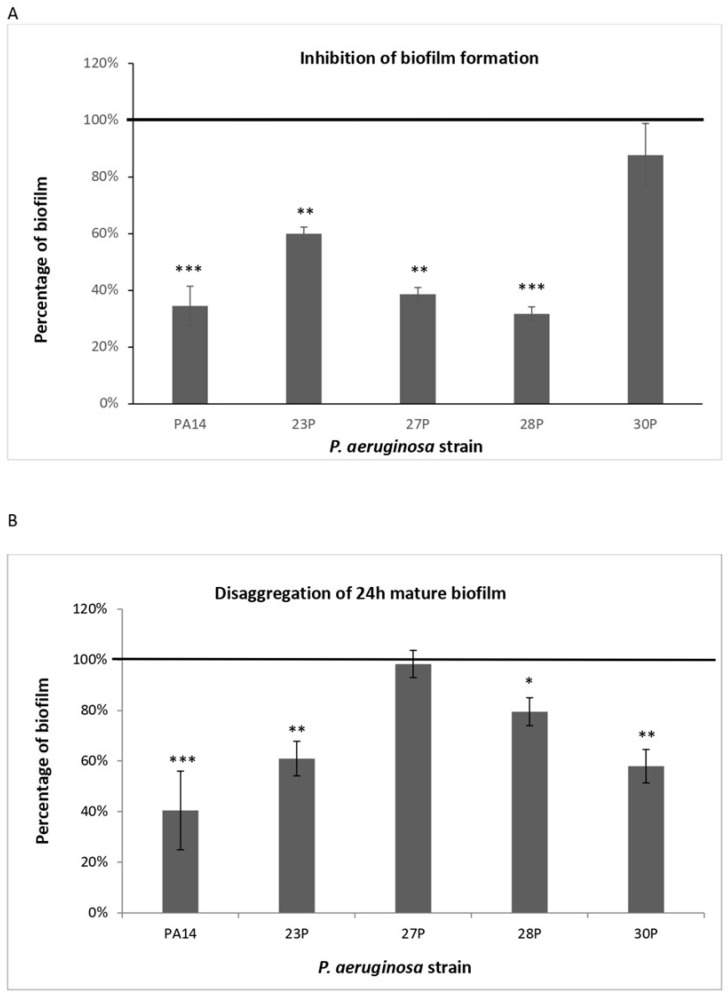
Effect of SPEP on biofilm formation of different clinical and reference strains. Panel (**A**): Effect of SPEP on biofilm formation. In the ordinate axis is reported the percentage of bacterial biofilm production. Data are expressed as the percentage of biofilm formed in presence of 200 U/mL SPEP compared with the untreated bacteria. Each data point is composed of 4 independent experiments, each performed at least in 3 replicates. Panel (**B**): Effect of SPEP on a 24 h mature biofilm. In the ordinate axis is reported the percentage of residual biofilm. Data are expressed as the percentage of residual biofilm after 24 h of treatment compared with the control sample. Each data point is composed of 4 independent experiments, each performed at least in 3 replicates. Error bars indicate the standard deviations of all the measurements. Statistical differences were determined by the Student’s *t*-test: * *p* < 0.05; ** *p* < 0.01, *** *p* < 0.001, compared with the control.

**Figure 3 ijms-23-12645-f003:**
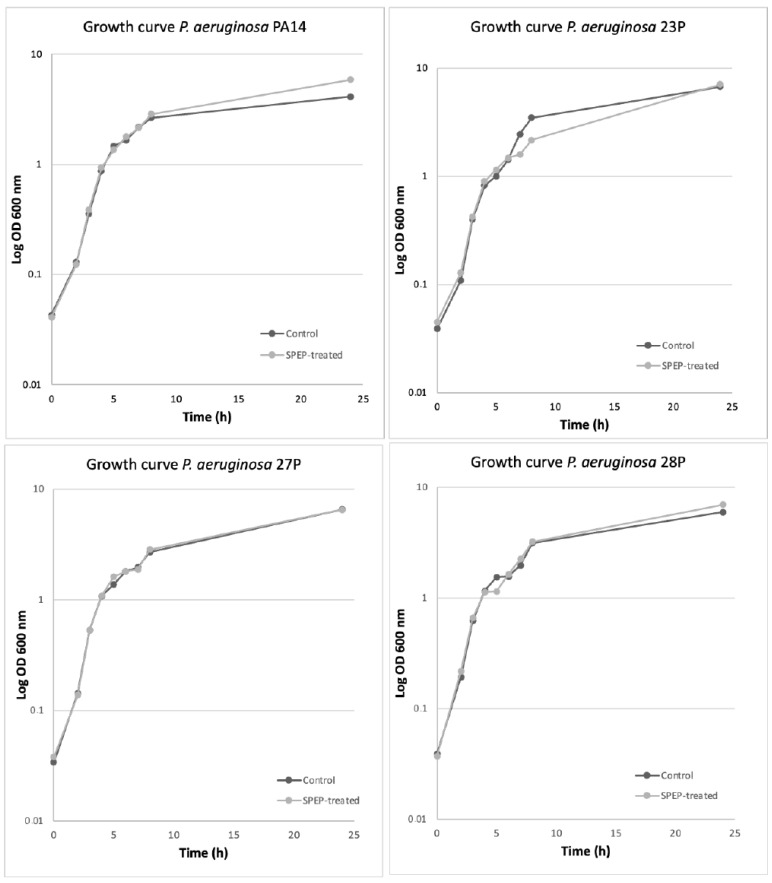
Growth curves of *P. aeruginosa* bacterial strains in planktonic conditions at 37 °C under orbital shaking (180 rpm) in the presence and absence of 200 U/mL SPEP. Curves are representative of three independent experiments.

**Figure 4 ijms-23-12645-f004:**
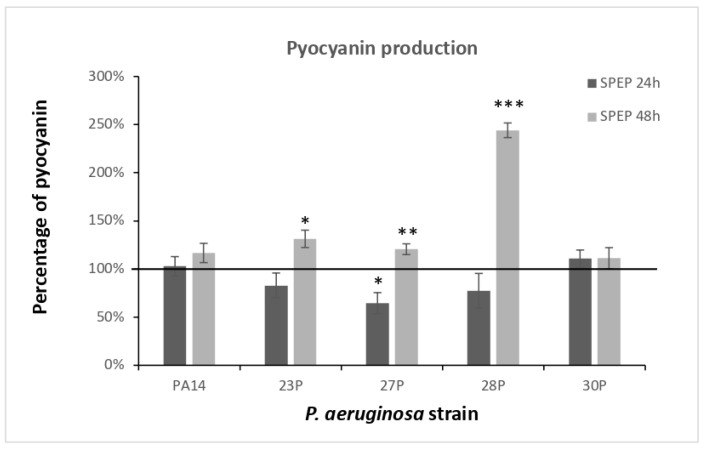
Effect of SPEP on pyocyanin produced after 24 and 48 h of growth. Data are reported as a percentage of pyocyanin production in the presence of SPEP in comparison with controls (bacteria grown in BHI). Each data point is composed of 4 independent experiments, each performed in at least 3 replicates. Error bars indicate the standard deviations of all the measurements. Statistica differences were determined by the Student’s *t*-test: * *p* < 0.05; ** *p* < 0.01, *** *p* < 0.001, compared with the control.

**Figure 5 ijms-23-12645-f005:**
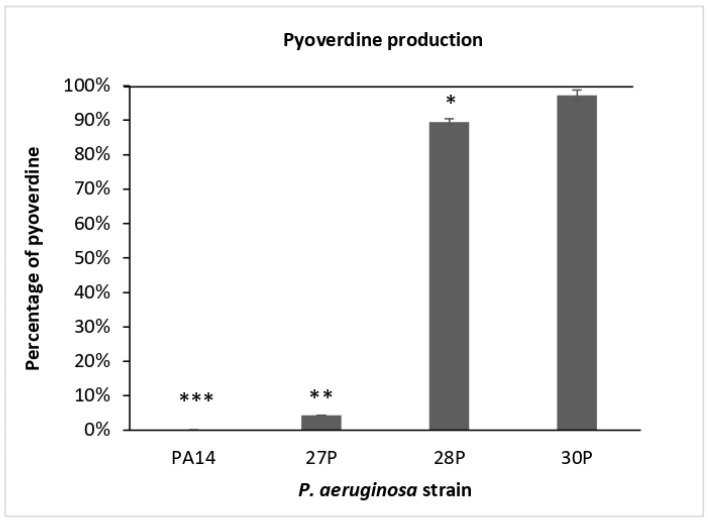
Effect of SPEP on pyoverdine produced after 48 h of growth. Data are reported as a percentage of pyoverdine production in the presence of SPEP in comparison with controls (bacteria grown in King’s B Medium supplemented with 0.5% (*w/v*) of CAA). Each data point is composed of 4 independent experiments, each performed in at least 3 replicates. Error bars indicate the standard deviations of all the measurements. Statistical differences were determined by the Student’s *t*-test: * *p* < 0.05; ** *p* < 0.01, ****p* < 0.001, compared with the control.

**Figure 6 ijms-23-12645-f006:**
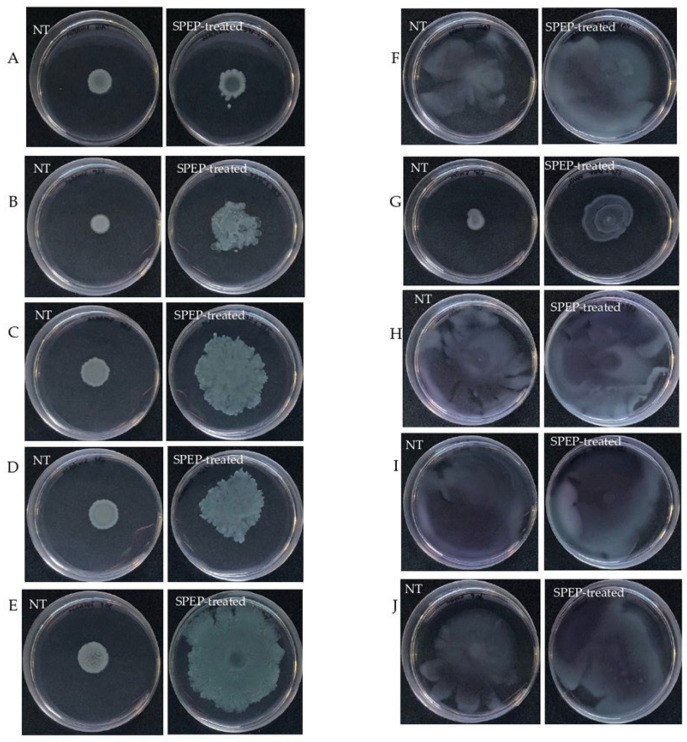
Motility assay (Petri plates diameter 8.5). Left Panel; Swarming assay of reference and clinical strains treated and not treated (NT) with SPEP. (**A**) *P. aeruginosa* PA14; (**B**) 23P; (**C**) 27P; (**D**) 28P; (**E**) 30P. Right Panel; Swimming assay of reference and clinical strains treated and not treated (NT) with SPEP. (**F**) *P. aeruginosa* PA14; (**G**) 23P; (**H**) 27P; (**I**) 28P; (**J**) 30P.

**Figure 7 ijms-23-12645-f007:**
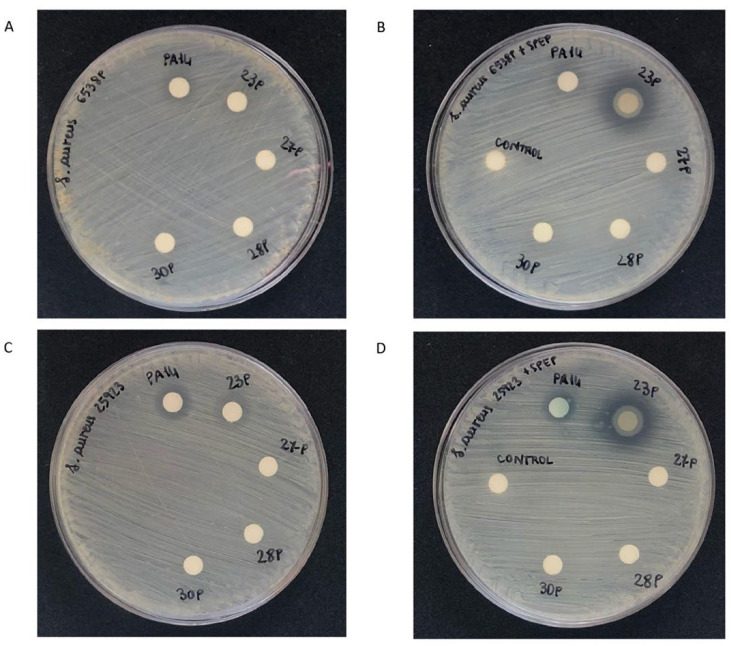
Disk diffusion susceptibility assay of *S. aureus* growth inhibition by the supernatant of *P. aeruginosa* treated and untreated with SPEP (Petri plates diameter 8.5). (**A**) Disk diffusion susceptibility assay of *S. aureus* 6538P containing supernatants derived from *P. aeruginosa* cultured in BHI. (**B**) Disk diffusion susceptibility assay of *S. aureus* 6538P containing supernatants obtained from *P. aeruginosa* cultured in BHI and 200 U/mL SPEP. (**C**) Disk diffusion susceptibility assay of *S. aureus* 25,923 containing supernatants derived from *P. aeruginosa* cultured in BHI. (**D**): Disk diffusion susceptibility assay of *S. aureus* 25,923 containing supernatants derived from *P. aeruginosa* cultured in BHI and 200 U/mL SPEP. Control disk contains 200 U/mL SPEP.

**Table 1 ijms-23-12645-t001:** Adhesion and invasion abilities of *P. aeruginosa* on A549 cells in the presence and in the absence of SPEP.

	Untreated		SPEP-Treated	
	Adhesion	Invasion	Adhesion	Invasion
*P. aeruginosa* PA14	1.41 × 10^5^ ± 0.08 × 10^5^	2.82 × 10^3^ ± 0.46 × 10^3^	1.16 × 10^5^ ± 0.21 × 10^5^	2.0 × 10^1^ ± 0.3 × 10^1^
*P. aeruginosa* 27P	3.04 × 10^5^ ± 0.21 × 10^5^	3.20 × 10^3^ ± 0.08 × 10^3^	1.79 × 10^5^ ± 0.45 × 10^5^	3.0 × 10^1^ ± 0.6 × 10^1^

Adhesion is expressed as CFU of bacteria that adhered to A549 cells 1 h post-infection at 37 °C. Invasion efficiency is expressed as CFU of bacteria that were gentamicin-resistant 1 h post-infection. Data represent the mean ± SD of three independent experiments.

**Table 2 ijms-23-12645-t002:** Phenotypic features of clinical and PA14 strains.

Bacterial Strain	Pyocyanin (OD 520 nm)	Pyoverdine (OD 460/400 nm)	Biofilm 24 h ^a^ (OD 590 nm)	Biofilm 48 h ^b^ (OD 590 nm)
PA14	0.178 ± 0.042	1821 ± 19	3.561 ± 0.357	13.470 ± 1.403
23P	0.148 ± 0.024	Not producer	3.175 ± 0.851	0.738 ± 0.373
27P	0.102 ± 0.005	765 ± 11	1.429 ± 0.643	3.049 ± 0.796
28P	0.093 ± 0.032	4871 ± 39	0.265 ± 0.038	2.540 ± 0.799
30P	0.115 ± 0.022	23740 ± 372	0.866 ± 0.345	5.013 ± 1.391

^a^ Biofilm production during a growth of 24 h without medium replacement. ^b^ Biofilm production after 48 h with medium replacement after 24 h.

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
