# Peer review of "Serratiopeptidase Affects the Physiology of Pseudomonas aeruginosa Isolates from Cystic Fibrosis Patients"

_ijms, 2022, doi:10.3390/ijms232012645_

Round 1

Reviewer 1 Report

The effect of serratiopeptidase treatment on Pseudomonas aeruginosa virulence is being investigated here. Different CF clinical isolates as well as the control PA14 strain are compared in this study for the effect of SPEP on the formation of biofilms, pyocyanin, pyoverdine, anti Staphylococcus activity (staphylolysin), motility and adhesion and cytotoxicity on eukaryotice cells. Although some results are obtained and sometimes puzzling, the effects of SPEP far from convincing.

Introduction

-PA14 does not have a mutation in RetS, but well in LadS!

- There are more references about the effects of pyocyanin and pyoverdine on virulence!

Results

- Figure 1, the effect of SPEP on biofilms formation is not very strong and not dose dependent. 

- Figure 4: SPEP seems to increase pyocyanin production, which seems to invalidate the claim that itcould impair virulence.

- PA14 does produce pyoverdine, only the BHI medium is not the medium of choice for its production since it contains sufficient iron to repress its production. Other media are the King's B medium of the CAA medium.

Reviewer 2 Report

The manuscript, “Serratiopeptidase affects adhesive features of Pseudomonas aeruginosa isolates from cystic fibrosis patients on biotic and abiotic substrates” describes the role of the metalloprotease serratiopeptidase (SPEP) on the virulence properties/factors of P. aeruginosa.  Factors measured include biofilm formation, biofilm dispersion, cell invasion, swarming motility, pyocyanin production, pyoverdine production, and staphylolytic protease activity.  The investigators used several clinical strains of P. aeruginosa in addition to the well characterized PA14 strain.  The use of several different strains was a bonus since it showed that the SPEP did not have the same effect on all strains of P. aeruginosa.  The investigators showed that SPEP had a moderate inhibitory effect on biofilms of PA14 and of several of the clinical strains.  They also showed that the SPEP can help disaggregate biofilms of some of the clinical strains.  They showed that the SPEP results in an increase in pyocyanin production at the 48 hr time point.   The SPEP had a varied effect on pyoverdine production, resulting in an increase in pyoverdine production in strain 28P but a decrease in pyoverdine in strain 30P.  The SPEP resulted in an increase in swarming motility for most strains.  The SPEP caused an increase in the stapholytic protease production in one of the strain (23P).

The SPEP definitely has physiological effects on P. aeruginosa, which is interesting.  However, the effects often seem to cause a significant increase in virulence factors, rather than a decrease (so likely it would not be a good infectious disease treatment strategy).  The investigators seems to focus on the inhibition of biofilm formation (which I would consider minor, since inhibition is at most 2-fold), and not focus on the significant effects of SPEP such as the increase in swarming, pyocyanin, and staphylolytic  protease production (these phenomena are not mentioned in the title or the abstract).  So, in my opinion, the SPEP is interesting in that it affects P. aeruginosa – it just happens to not do what was predicted.  Why not rewrite the paper emphasizing that SPEP affects P. aeruginosa physiology, even though it would not make a good treatment strategy?

Other comments for improving the manuscript:

1) The abstract is written more as an introduction than as an abstract.  The intro part of the abstract can be reduced and the results can be emphasized (swarming, pyocyanin, pyoverdine are not mentioned in the abstract).

2) The authors state that the SPEP causes a “dose-dependent effect on PA14 biofilm formation”.  The data in Figure 1 do not show a dose-dependent response.  The response looks random to me.  Can a linear model be drawn with dose and response?

3) The authors state that “We could speculate that SPEP favors a transition from the sessile to  the planktonic phenotype, more easily recognizable by the immune system.”  However, I would not consider the effect of SPEP on swarming motility a planktonic phenotype.  Swarming motility is associated with surfaces.  I didn’t see data on swimming motility.

4) In the discussion the investigators state that, “Therefore, in general serratiopeptidase could be considered as a strategy of choice against P. aeruginosa  infections difficult to eradicate with conventional antibiotics.”  I suppose it could be considered, but generally a treatment that increases virulence factor production would not make for a good treatment strategy.

Round 2

Reviewer 1 Report

The authors did their best to revise the manuscript and, although the results are sometime divergent depending on the clinical isolate investigated, the take home message here is that one should include different isolates before taking conclusions. It will therefore be interesting to investigate why different strains react sometimes in opposing ways. 
